# Minding the margins: Evaluating the impact of COVID-19 among Latinx and Black communities with optimal qualitative serological assessment tools

**Raquel A. Binder**[1◉]*, **Angela M. Matta**[1◉], **Catherine S. Forconi**[1], **Cliff I. Oduor**[2], **Prajakta Bedekar**[3,4], **Paul N. Patrone**[3], **Anthony J. Kearsley**[3], **Boaz Odwar**[1], **Jennifer Batista**[5], **Sarah N. Forrester**[5], **Heidi K. Leftwich**[6], **Lisa A. Cavacini**[1], **Ann M. Moormann**[1]

1 Department of Medicine, University of Massachusetts Chan Medical School, Worcester, MA, United States of America, 2 Department of Pathology and Laboratory Medicine, Warren Alpert Medical School, Brown University, Providence, RI, United States of America, 3 Applied and Computational Mathematics Division, National Institute of Standards and Technology, Gaithersburg, MD, United States of America, 4 Department of Applied Mathematics and Statistics, Johns Hopkins University, Baltimore, MD, United States of America, 5 Department of Population and Quantitative Health Sciences, University of Massachusetts Chan Medical School, Worcester, MA, United States of America, 6 Department of Obstetrics and Gynecology, University of Massachusetts Chan Medical School, Worcester, MA, United States of America

◉ These authors contributed equally to this work.
* raquel.binder@umassmed.edu

**Data Availability Statement:** All data are in the manuscript and/or supporting information files. Hence, the complete and clean data set with all the

## Abstract

COVID-19 disproportionately affected minorities, while research barriers to engage underserved communities persist. Serological studies reveal infection and vaccination histories within these communities, however lack of consensus on downstream evaluation methods impede meta-analyses and dampen the broader public health impact. To reveal the impact of COVID-19 and vaccine uptake among diverse communities and to develop rigorous serological downstream evaluation methods, we engaged racial and ethnic minorities in Massachusetts in a cross-sectional study (April—July 2022), screened blood and saliva for SARS-CoV-2 and human endemic coronavirus (hCoV) antibodies by bead-based multiplex assay and point-of-care (POC) test and developed across-plate normalization and classification boundary methods for optimal qualitative serological assessments. Among 290 participants, 91.4% reported receiving at least one dose of a COVID-19 vaccine, while 41.7% reported past SARS-CoV-2 infections, which was confirmed by POC- and multiplex-based saliva and blood IgG seroprevalences. We found significant differences in antigen-specific IgA and IgG antibody outcomes and indication of cross-reactivity with hCoV OC43. Finally, 26.5% of participants reported lingering COVID-19 symptoms, mostly middle-aged Latinas. Hence, prolonged COVID-19 symptoms were common among our underserved population and require public health attention, despite high COVID-19 vaccine uptake. Saliva served as a less-invasive sample-type for IgG-based serosurveys and hCoV cross-reactivity needed to be evaluated for reliable SARS-CoV-2 serosurvey results. The use of the developed rigorous

relevant metadata has been submitted and will be published as supplemental information.

**Funding:** This work was supported through the National Institutes of Health, NCI Serological Sciences Network (U01 CA261276), UMass Chan COVID-19 pandemic research fund, MassCPR Evergrande Award, and the National Center for Advancing Translational Sciences, National Institutes of Health, through Grant KL2-TR001455. The content is solely the responsibility of the authors and does not necessarily represent the official views of the NIH. The funders had no role in study design, data collection and analysis, decision to publish, or preparation of the manuscript.

**Competing interests:** The authors have declared that no competing interests exist.

downstream qualitative serological assessment methods will help standardize serosurvey outcomes and meta-analyses for future serosurveys beyond SARS-CoV-2.

## Introduction

Differential health care access and exposure risks have led to racial and ethnic COVID-19 disparities in the United States (US), leaving Latinx and Black communities to experience disproportionately high SARS-CoV-2 infection rates and COVID-19 related morbidity and mortality [1, 2]. Seroprevalence studies have become essential public health tools to assess the regional spread and pre-existing immunity to SARS-CoV-2 among at-risk populations [3–5]. Further, by distinguishing between anti-SARS-CoV-2 spike (S), receptor-binding domain (RBD), and nucleocapsid (N) antibodies, COVID-19 vaccine uptake (RBD/S only) can be estimated and compared to past SARS-CoV-2 infections [6]. Anti-SARS-CoV-2 immunoglobulin (Ig)G and IgA antibodies can be measured in both blood and saliva, the latter serving as less invasive sample collection alternative [7].

Here, we engaged ethnic and racial minorities to evaluate the impact of COVID-19 in the Greater Worcester area from April to July of 2022 and assessed blood- and saliva-based serosurvey methods. Overall, this study (i) evaluated the impact of COVID-19 and vaccine uptake among marginalized communities, (ii) confirmed the utility of using saliva for serosurveys, (iii) compared the utility of a bead-based multiplex assay vs. a point-of-care (POC) test for SARS-CoV-2 antibody measurements, and (iv) demonstrated the benefit of developing and using classification boundary methods for optimal interpretation of serological assays.

## Methods

### Participant recruitment and sample collection

This cross-sectional study was approved by the University of Massachusetts Chan Medical School (UMass Chan) Institutional Review Board (IRB Docket # H00023083). Structured interviews with Black and Latinx community members in the Greater Worcester Area in Massachusetts (MA) were conducted to identify recruitment barriers for participation in research studies. IRB-approved study flyers were distributed prior to engagement with local stakeholders. We joined regular community gatherings organized by Net of Compassion, Centro Hispano, and Central MA YMCA in Worcester, along with the St. John Catholic Church in Clinton for in-person recruitment events implemented both in Spanish and English from April 21st to July 4th of 2022. On site, we provided a fact sheet explaining the purpose of the study and were available to answer questions in Spanish and English. We obtained informed verbal consent from eligible individuals (inclusion criteria: 18+ years of age, exclusion criteria: prisoners and people unable to communicate in English or Spanish) and participants were asked to fill out a brief RedCap survey covering demographic information, COVID-19 vaccine, and SARS-CoV-2 infection history through tablets that were provided by the study team or by QR codes that could be scanned on personal devices. We used verbal (not written) consent for this minimal risk study due to hesitation of documentation and decreased literacy rate among individuals from underserved communities. The verbal consent was documented by the research team member interviewing the participant and this method was approved by the IRB prior to study begin. Blood and saliva samples were collected with Tasso SST devices (Tasso, Inc., Seattle, WA) and SuperSal2® devices (Oasis Diagnostics, Vancouver, WA), respectively, as per manufacturer's guidelines.

SARS-CoV-2 anti-immunoglobulin (Ig)G and IgM antibodies were measured with an emergency use authorized approved POC test (FaStep from Assure Tech, Hangzhou, China), which detects both anti-SARS-CoV-2 N and S antibodies. The POC test results were provided to participants immediately, along with a $50 reimbursement. See Supplemental Methods for more information on sample collection and processing.

## Multiplex Luminex assay

The following SARS-CoV-2 antigens were coupled to Luminex MagPlex Microspheres as indicated by the manufacturer: Wild-type (WT; Wuhan) full-length spike (S), WT nucleocapsid (N), WT receptor-binding domain (RBD), RBD alpha, RBD beta, RBD gamma, RBD delta, RBD lambda, and RBD omicron. Following human endemic coronavirus (hCoV) antigens and a Bovine Serum Albumin (BSA) control were coupled: HKU1, OC43, NL63, and 229E, see Supplemental Methods for more details. Briefly, after validation of conjugated beads, the participant samples were screened on 96- or 384-well plates, including a seven- or ten-point serial dilution standard. Conjugated beads covering the antigen panel were combined and washed, incubated, washed again, and biotinylated anti-human secondary IgG or IgA (BD Pharmingen) antibody added. After incubation and washing, phycoerythrin conjugated streptavidin was added (BD Pharmingen). Finally, after incubation, washing, and resuspension, the plate was screened by a FlexMap 3D Luminex instrument. The antigen-specific median fluorescence intensity (MFI) for each sample was recorded and BSA subtracted (including for the standards) to account for non-specific bead binding [8]. Saliva samples were screened for total IgG and total IgA antibodies to account for differential salivation flow rates by coupling anti-human IgG gamma chain (Bio-Rad, Hercules, CA) and anti-human IgA alpha chain protein (Abcam, Cambridge, UK) respectively.

Previously described de-identified banked blood samples served as positive (n = 50) and negative blood controls (n = 50) [9]. Banked saliva samples (n = 50), collected with the same SuperSal2 devices, served as alternative control group for the seroprevalence calculation, see "Qualitative Serological Assessments" section below.

## Across-plate normalization

Dilution series of standards for each antigen-plate combination were weighted using a plate-dependent variance while a normalization factor was computed with custom MATLAB scripts. This factor was then applied by multiplying it with each antigen/isotype-specific sample MFI on the plate, see Supplemental Methods for more details.

## Qualitative serological assessments

Sample MFIs were translated to qualitative (i.e., binary positive/negative) outcomes as described in reference [10] and in the Supplemental Methods. Briefly, for the blood samples, empirical training data were taken as approximate probability models of measurement outcomes for each antigen, conditioned on knowing the class of the underlying sample. The analysis was applied to multidimensional data by treating up to three antigens as distinct axes in a coordinate space, see **S1 Fig** for a three-dimensional analysis example. Due to the lack of collection method- and population-matched controls, the saliva-based IgG seroprevalence calculations were determined with alternative control samples from a 2020 Kenyan study (proxy for negative training data), see Supplemental Methods for more details.

## Statistical analysis

Statistical calculations and graphs were done in Prism v9.4.1, R v2023.09.1+494, and MATLAB R2023a Update 5 (9.14.0.2337262).

## Results

### Demographics and vaccine/infection history

A total of 290 adult study participants were enrolled in Worcester, Shrewsbury, and Clinton, MA between April and July of 2022. Most participants donated blood (98.6%, n = 286) and saliva (94.8%, n = 275) samples. Participants who did not give a saliva sample mostly lacked saliva production (i.e., dry mouth), especially among elderly, but were not hesitant to donate the sample. The demographic, clinical, and SARS-CoV-2 infection history survey was filled out by 98.3% participants (n = 285), while 47.6% (n = 138) chose to answer in Spanish and 51.7% (n = 150) in English (see Supplemental Materials for full survey). Most participants self-identified as Latinx/Hispanic (67.6%, n = 196), and female (61.0%, n = 177), while the average age was 45 years (range: 18–82, STDV: 16.3), and 31.4% (n = 91) of participants received a college or higher education degree, see **Table 1**. Non-Hispanic White participants constituted 13.4% (n = 39) of the study population. Among the participants, 15.5% (n = 45) reported pre-existing health conditions, most commonly hypertension, obesity, diabetes type II, and asthma, while 13.5% (n = 39) reported smoking or vaping prior to the pandemic, see **S1 Table** in S1 File. Of note, the self-reported pre-existing health conditions (particularly the high prevalence of hypertension, diabetes, and asthma) reflect health-related risk factors reported among US minorities in other studies and increase the vulnerability to COVID-19 associated complications [11–13].

A total of 265 (91.4%) participants received at least one dose of a COVID-19 vaccine; 45.7% (n = 121) Moderna; 40.0% (n = 106) Pfizer; 8.6% (n = 23) Johnson and Johnson; and 5.7% (n = 15) 'other', for the base vaccine series, see **S1 Table** in S1 File. The vaccine uptake was lower for the second dose (84.5%, n = 245) and booster (68.3%, n = 198) while most participants received an influenza vaccine in the past 5 years (79.3%, n = 230). As for vaccine-related side effects, 37.7% (n = 100), 43.7% (n = 107), and 35.9% (n = 71) of participants did <u>not</u> experience any post-COVID-19 vaccine symptoms post-first dose, -second dose and -booster, respectively. Among those who experienced vaccine-associated symptoms post-base vaccine series and booster, the average severity scores were 4.1 (STDV: 2.4; range: 1–10) and 3.6 (STDV: 2.2; range:1–10) respectively on a scale from 1 to 10, while arm soreness, fever, and fatigue were the most frequently reported symptoms. Further, 16.6% (n = 44), 14.7% (n = 36), and 6.1% (n = 12) of participants experienced severe symptoms (rating $\geq$ 6) post-first dose, second dose, and booster, respectively, encompassing thrombosis, strokes, fainting, and migraines. A total of 6 participants reported vaccine-associated hospitalizations, encompassing all doses for all participants. Among those who were not vaccinated, participants reported being hesitant because of fear of COVID-19 vaccine side effects, lack of trust in the vaccine, and not knowing enough about the vaccine.

A total of 121 (41.7%) participants reported testing positive for SARS-CoV-2 at least once and the associated average symptom severity score was 5.4 (STDV: 2.4; range: 1–10), while 43.0% (n = 52) experienced severe symptoms (rating $\geq$ 6) and 9.1% (n = 11) were hospitalized. The most common symptoms were body aches (52.1%, n = 63), fever (51.2%, n = 62), fatigue (51.2%, n = 62), cough (48.8%, n = 59), headache (46.3%, n = 56), sore throat (45.5%, n = 55), congestion/runny nose (38.0%, n = 46), and loss of smell or taste (36.4%, n = 44). A total of 7 individuals (5.8%) reported not having any symptoms associated with the positive test (i.e.,

**Table 1. Study participant demographics from 290 study participants enrolled in Massachusetts from April to July 2022 and 286 associated blood samples collected during the study period.**

| | Overall | Nucleocapsid IgG *Positive* | Nucleocapsid IgG *Negative* | $\chi^2$ |
|---|---|---|---|---|
| | | Serum | Serum | |
| | n (%) | n (%) | n (%) | p-value[∋] |
| Total | 290 (100) | 136 (49.9±7)[#] | 150 (52.5) | |
| **Age** (years) | | | | |
| 18–40 | 126 (43.5) | 52 (38.2) | 72 (48.0) | 0.292 |
| 41–60 | 96 (33.1) | 45 (33.1) | 51 (34.0) | |
| 60 | 59 (20.3) | 31 (22.8) | 26 (17.3) | |
| Missing | 9 (3.1) | 8 (5.9) | 1 (0.7) | |
| **Gender** | | | | |
| Female | 177 (61.0) | 79 (58.1) | 96 (64.0) | 0.393 |
| Male | 109 (37.6) | 54 (39.7) | 53 (35.3) | |
| Non-binary | 3 (1.0) | 2 (1.5) | 1 (0.7) | |
| Missing | 1 (0.4) | 1 (0.7) | 0 (0.0) | |
| **Race** | | | | |
| White | 97 (33.5) | 39 (28.7) | 56 (37.3) | 0.039* |
| Mixed/Mestizo | 90 (31.0) | 53 (39.0) | 37 (24.7) | |
| Black/African American | 29 (10.0) | 14 (10.3) | 15 (10.0) | |
| Asian | 31 (10.7) | 9 (6.6) | 21 (14.0) | |
| Other | 43 (14.8) | 21 (15.4) | 21 (14.0) | |
| **Ethnicity** | | | | |
| Hispanic | 196 (67.6) | 104 (76.5) | 90 (60.0) | 0.0013** |
| non-Hispanic | 91 (31.4) | 29 (21.3) | 60 (40.0) | |
| Missing | 3 (1.0) | 3 (2.2) | 0 (0.0) | |
| **Education** | | | | |
| High School or less | 100 (34.5) | 57 (41.9) | 40 (26.7) | 0.0193* |
| College or more | 91 (31.4) | 37 (27.2) | 54 (36.0) | |
| Missing | 99 (34.1) | 42 (30.9) | 56 (37.3) | |
| **SARS-CoV-2 + Test** | | | | |
| Yes | 121 (41.7) | 71 (52.2) | 49 (32.7) | 0.0003*** |
| No | 163 (56.2) | 59 (43.4) | 101 (67.3) | |
| Missing | 6 (2.1) | 6 (4.4) | 0 (0.0) | |
| **Fully Recovered**[ϕ] | | | | |
| Yes | 108 (37.2) | 54 (39.7) | 53 (35.3) | 0.0099** |
| No | 21 (7.2) | 18 (13.3) | 3 (2.0) | |
| Long COVID | 19 (6.6) | 12 (8.8) | 7 (4.7) | |
| Never had COVID | 136 (46.9) | 46 (33.8) | 87 (58.0) | |
| Missing | 6 (2.1) | 6 (4.4) | 0 (0.0) | |

[∋]The "Missing" categories, the "Non-binary" category in gender, and the "Never had COVID" category under "Fully Recovered" were omitted for the $\chi^2$ test comparison; p-value < 0.05. The gender, ethnicity, education, and SARS-CoV-2 test comparison were done with the Fisher's exact test (2x2 categories). * p-value < 0.05, ** p-value < 0.01, ***p-value < 0.001.

[#]136 is the empirical count of positives (equal to a 47.5% empirical seroprevalence) and 49.9% (95% CI: ±7) is the calculated bias-corrected seroprevalence.

[ϕ]COVID-19 recovery among all study participants (including 121 test-confirmed cases). Self-reporting having fully recovered from a SARS-CoV-2 infections ("Yes") or not ("No"), whether infection-associated symptoms lasted past 4 weeks of the initial infection and therefore reported long COVID-19 symptoms (Long COVID), and those who never had COVID-19 (Never had COVID). For the $\chi^2$ test comparison the "Missing" and the "Never had COVID" categories were omitted.

asymptomatic infections). There was an association between reporting pre-existing health conditions and elevated severity scores (rating ≥ 6, Fisher's exact test, p = 0.0475) but not between smoking/vaping and elevated severity scores (rating ≥ 6, Fisher's exact test, p = >0.999). Among those with confirmed SARS-CoV-2 infections (n = 121), 26.5% (n = 32) reported not having fully recovered from the infection and 14.1% (n = 17) reported long COVID-19 symptoms (persistent symptoms past 4 weeks of diagnosis). For further analysis, the categories "not fully recovered from the infection" and reporting "long COVID-19 symptoms" were collapsed. Within this category, 78.1% (n = 25) were female, 87.5% (n = 28) were Latinx, 59.4% (n = 19) of mixed/mestizo race, and the average age was 48.6 years (STDV: 16.4, range: 19–82). There was a significant association between being 50 + years of age and not having fully recovered from COVID-19 symptoms (Fisher's exact test, p = 0.0135). There was no significant correlation between being female or reporting pre-existing health conditions and not having recovered from COVID-19 symptoms. Since the majority of the study population self-identified as Latinx and/or belonging to racial minorities, we were not able to contrast unresolved COVID-19 symptoms or any other variables among racial/ethnic sub-groups.

## Blood-based SARS-CoV-2 antibodies

Among those who received a SARS-CoV-2 POC antibody results (n = 284), 93.7% (n = 266) were positive for IgG and 4.6% (n = 13) for IgM. The POC test covered both SARS-CoV-2 N and S antibodies together, while the multiplex assay allowed measuring presence of antibodies based on individual antigens and therefore distinguish between vaccine- (S/RBD-only since all currently FDA-approved COVID-19 vaccines in the US are S/RBD-based and do not include the N antigen) and infection-induced antibodies. The serum multiplex-based *IgG* seroprevalences resulted in 97.5% ± 2.4% (95% CI) for RBD/S/N (3 antigens), 99.9% ± 3.4% (95% CI) for RBD/S (2 antigens), 97.2% ± 2.0% (95% CI) for S/N (2 antigens), 96.5% ± 2.2% (95% CI) for RBD/N (2 antigens), 96.5% ± 2.2% (95% CI) for RBD, and 97.9% ± 1.7% (95% CI) for S (see **S2 Table** in S1 File), mirroring the high self-reported vaccination uptake. The serum-based IgG N seroprevalence was 49.9% ± 7.0% (95% CI), indicating past infection rather than vaccination, similar to the percent of self-reported exposures. Statistically significant differences in N-specific serological test outcomes were observed among study participants based on race, ethnicity, education, and having fully recovered from COVID-19 symptom, as well as based on self-reported SARS-CoV-2 test results prior to study participation, see **Table 1**. The S- and RBD-specific serological test outcomes were not contrasted based on the listed categories due to the high overall seroprevalence and self-reported vaccine uptake being close to 100%. As for SARS-CoV-2 variants, the delta variant had the highest mean MFI and reached the highest maximum MFI read, see **Fig 1**. While antigen-specific MFI are influenced by antigen quality and steric hindrance this may reflect that SARS-CoV-2 delta variant antibodies were the most abundant among our study population at the time of sample collection.

The concordance between the POC test results and qualitative multiplex assay outcomes was high. Accordingly, 94.0% (267/284) of the outcomes aligned between the POC test and the three antigen RBD/S/N readouts (i.e.,17 outcomes did not match: 14 were positive by multiplex [RBD, S and N] but negative by POC, and 3 were negative by multiplex but positive by POC). Similarly, 94.4% (268/284) of the outcomes aligned between POC and the two antigen S/N analysis, 94.4% (268/284) of the outcomes aligned between POC and the two antigen RBD/N analysis, and 93.7% (266/284) of the outcomes aligned between POC and the two antigen RBD/S analysis. Comparing the POC outcomes with antigen specific MFIs revealed a correlation between a positive POC test and increasing MFI multiplex assay measurements for S

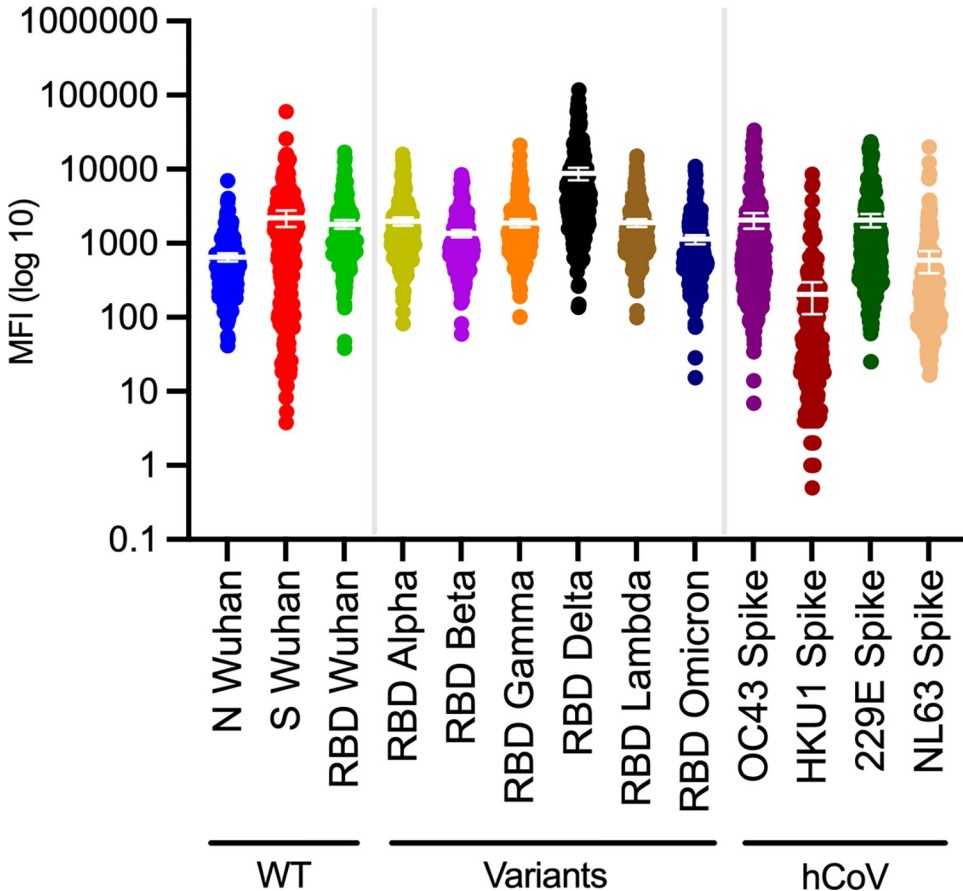

**Fig 1. Serum IgG outcome distribution.** Dot plot with means and 95% confidence intervals (CI) of antigen-specific outcomes (MFI—BSA) for serum IgG, covering SARS-CoV-2 (including variants) and human endemic coronaviruses (OC43, HKU1, NL63, 229E) with log10 y-axis. See **S2 Fig** for plots with linear y-axis. MFI, median fluorescence intensity. N, Nucleocapsid. S, Spike. RBD, Receptor Binding Protein. hCoV, human endemic coronaviruses. WT, wildtype (Wuhan).

and RBD, but not for N (see **Fig 2**), indicating that the POC test was reliably detecting the RBD/S antibodies measured by the multiplex assay.

To analyze the potential cross-reactivity between SARS-CoV-2 and hCoVs, we evaluated IgG OC43, HKU1, 229E, and NL63 antibody levels in the blood samples. OC43 and 229E had the highest MFI outcomes, see **Fig 1**. Given that OC43 is closely related to SARS-CoV-2 (both β-CoV members) [14], we compared SARS-CoV-2 and OC43 antibody measurements (both S-based) and found the correlation to be low ($R^2 = 0.2$), see **Fig 3**. Still, the antibody measurements for both were high, see **Fig 1,** and a paired t-test did not find the paired OC43 and SARS-CoV-2 measurements to be significantly different, see **Fig 3**. Hence, cross-reactivity between IgG S measurements for SARS-CoV-2 and OC43 could not be ruled out. We further analyzed the samples that were non-concordant between the multiplex analysis and the POC test for cross-reactivity (i.e., samples that were positive for the multiplex assay and negative for the POC test across all antigen combinations). The average SARS-CoV-2 S MFI of the non-concordant samples (n = 15) was lower compared to the concordant samples (n = 265), but the S-based OC43 measurements were not significantly higher (see **S3 Fig**). Hence, the sensitive and specific (**S2** and **S3 Tables** in S1 File) multiplex assay was more likely to pick up a positive SARS-CoV-2 sample compared to the POC test but it was not more likely to pick up

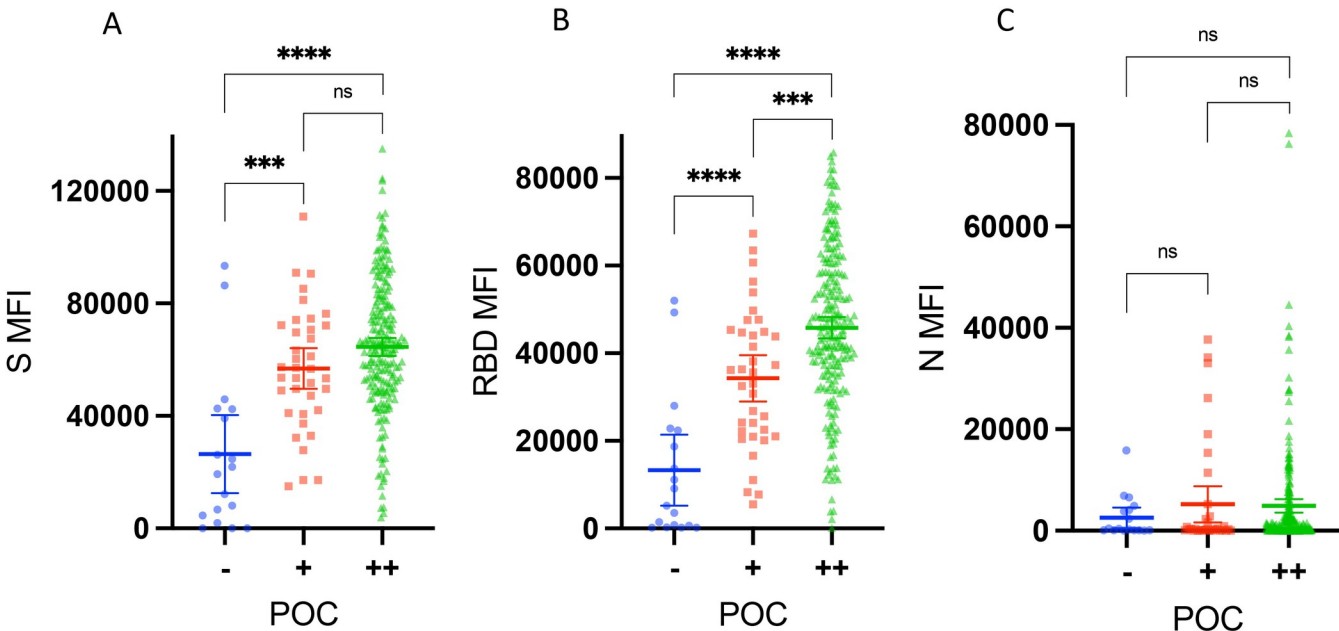

**Fig 2. Antigen-specific median fluorescence intensity read out broken down by point-of-care test outcome.** Comparing the point-of-care test outcome (POC) outcomes with antigen specific median fluorescence intensity (MFIs) revealed a correlation between a positive POC test and increasing MFI multiplex measurements for **(A)** S (ns, p = 0.06) and **(B)** RBD, but not for **(C)** N (ns, p = 0.19, p = 0.06, p = 0.86). MFI, median fluorescence intensity. N, Nucleocapsid. S, Spike. RBD, Receptor Binding Protein. POC, point-of-care test result. (-), POC negative. (+), POC positive (light band). (++), POC positive (dark band). Ns, non-significant (p≥0.05). *** = p<0.001. **** = p<0.0001.

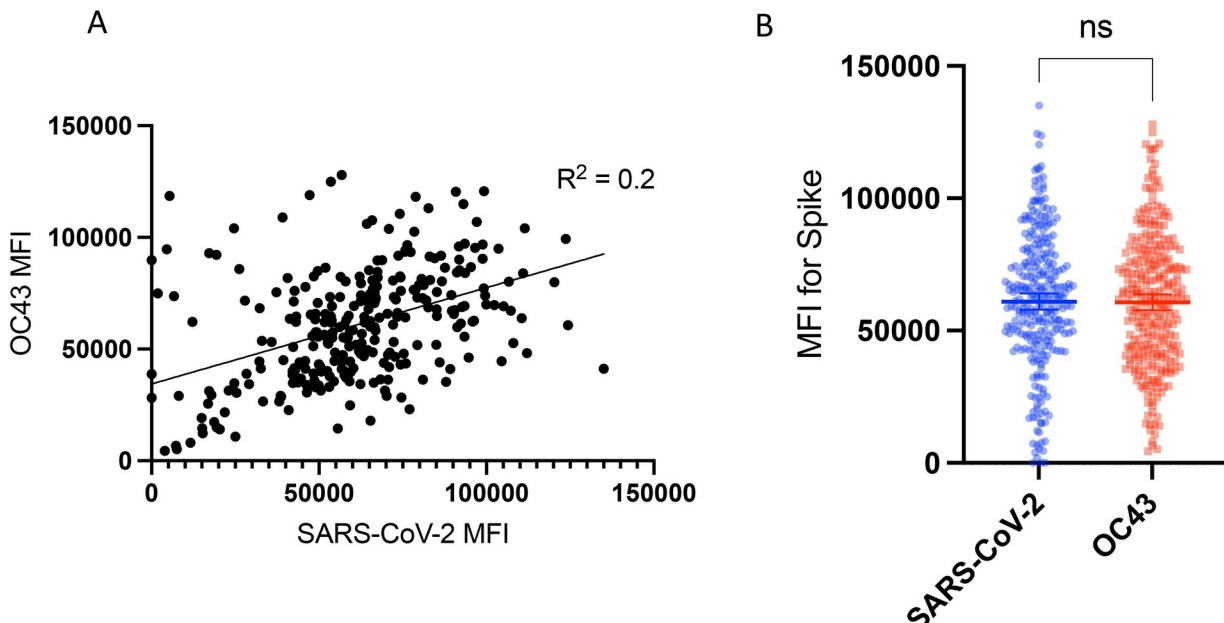

**Fig 3. Comparison of SARS-CoV-2 and OC43 antibody measurements. (A)** There was no/low correlation ($R^2$ = 0.2) between SARS-CoV-2 and OC43 S-based MFI, both Spike [S]-based MFI. **(B)** Still, the paired antibody measurements for OC43 and SARS-CoV-2 were not significantly different by paired t test (p = 0.84). Hence, cross-reactivity between IgG S measurements for SARS-CoV-2 and OC43 could not be ruled out. MFI, median fluorescence intensity. Ns, non-significant (p≥0.05).

OC43. Note that the statistical outcomes may be affected by the unequal sample sizes (15 vs 265).

While anti-SARS-CoV-2 IgA has been shown to wane faster than IgG, mucosal and blood-based IgA may provide protection from infections [7, 15–17]. Hence, we measured serum-based *IgA* seroprevalence which resulted in 87.2% ± 6.4% (95% CI) for RBD/S/N (3 antigens), 83.1% ± 6.7% (95% CI) for RBD/S (2 antigens), 84.8% ± 6.6% (95% CI) for S/N (2 antigens), 39.8% ± 12.2% (95% CI) for RBD/N (2 antigens), 62.7% ± 14.5% (95% CI) for RBD, 84.0% ± 6.7% (95% CI) for S, and 14.1% ± 25.5% (95% CI) for N, see **S3 Table** in S1 File. Comparing the antigen-specific outcomes in blood resulted in significantly lower MFIs for IgA compared to IgG for all antigens (p<0.0001; Welch's t test), see **S4, S5,** and **S6 Figs**.

### Saliva SARS-CoV-2 antibodies

The saliva-based *IgG* seroprevalences and uncertainty range (approximating the 100% confidence interval) resulted in 100.0% (98.7–100.0) for RBD/S/N (3 antigens), 100.0% (98.7–100.0) for RBD/S (2 antigens), 96.0% (92.4–99.6) for S/N (2 antigens), 86.9% (81.6–96.4) for RBD/N (2 antigens), 86.9% (75.8–96.2) for RBD, 96.0% (92.4–99.6) for S, and 48.0% (48.0–99.7) for N, see **S4 Table** in S1 File. The SARS-CoV-2 seroprevalences from saliva and serum resulted in comparable outcomes for IgG (see **S2** and **S4 Tables** in S1 File), and the concordance between the qualitative positive/negative antibody results was high; 97.6% (279/286) of the results aligned for the IgG RBD/S/N (3 antigen) analysis (i.e., 7 participants had positive saliva samples but negative blood samples by IgG RBD/S/N multiplex analysis). Note that the concordance reflects the underlying high overall seroprevalence. A direct comparison of saliva and blood MFIs was not possible since saliva antibody measurements (MFI minus BSA, divided by total Ig, and multiplied by 1000) were transformed differently than blood/serum (MFI minus BSA) to account for variation in salivary flow rates and comparing the differently transformed MFIs between serum and saliva for each SARS-CoV-2 antigen did not identify a correlation, see **S7 Fig**.

The saliva-based *IgA* seroprevalences could not be calculated because the antigen-specific outcomes (transformed MFIs) between the study participant and alternative control samples overlapped significantly, see **S8 Fig**. Hence, no classification boundaries and therefore no percent seroprevalence could be established. However, comparing the antigen-specific measurements (MFI) in saliva resulted in significantly lower reads for IgA compared to IgG for the RBD and S antigen outcomes (p<0.0001; Welch's t test), see **S9 and S10 Figs.** Whereas for the N-specific outcomes in saliva, there were less overall antibodies (e.g., lower transformed MFIs compared to RBD and S), and the IgA reads were significantly higher compared to IgG, see **S11 Fig**.

### Discussion

This study outlines an effective culturally sensitive recruitment method that overcame research study access barriers generally reported among US minority populations, enrolling 290 participants within four months [2, 18–20]. Among the mostly Latinx young to middle aged peri-urban MA study population, the majority reported being vaccinated (91.4%), which was confirmed by blood and saliva IgG antibody screening. According to the MA Department of Public Health, 86.9% of the MA population had received at least one dose of a COVID-19 vaccine by July 2022 [21] and 81.4% of the general US population had received at least one COVID-19 vaccine dose by May 2023 [22]. Hence, our diverse study population had a high vaccine uptake and did not reflect the reported vaccine hesitancy among minorities [19, 20, 23]. This may have been due to widely available COVID-19 vaccines in MA as vaccine availability and

general ease of access has been cited as one of the main uptake barriers among marginalized groups [19, 20].

The percent of self-reported infections aligned with the N-based IgG seroprevalence outcomes in serum (N: 49.9%) and saliva (N: 48.0%). Hence, within two years and four months (the first COVID-19 case in MA was confirmed on Feb 1[st], 2020 [24] and the study recruitment ended in July of 2022) about half our diverse study population had been exposed to SARS-CoV-2. While the percent of self-reported infections aligned with the N-based IgG serum and saliva seroprevalence results, we found that reporting a positive SARS-CoV-2 test was not necessarily linked to the presence of anti-N IgG antibodies (**Table 1**). This is likely due to the relatively short half-life of anti-N IgG antibodies. Others have found that anti-N antibodies start declining within one month post-positive PCR test with over half the study population testing seronegative within 6–7 months [25, 26]. Since our study was implemented 2 + years after the first local COVID-19 case and since antibody levels may range across individuals, it is likely that the antibody levels had dropped below the detection limits by the time we collected and screened the blood samples of these individuals. Similarly, we found that participants who did not report a confirmatory test were positive for anti-N IgG antibodies. Given that 4% to 41% of SARS-CoV-2 infections may be asymptomatic [27], it is likely that these individuals may have been infected but were not aware or did not seek testing.

The reported average severity scores were higher for infections (5.4) as compared to vaccination (4.1 and 3.6 for baseline doses and booster, respectively) and the overall number of vaccine-associated hospitalizations were lower (n = 6) compared to infection-associated hospitalizations (n = 11). Additionally, 26.5% of individuals who reported past infections had not fully recovered and 14.1% reported long COVID-19 symptoms. Among those who experienced lingering COVID-19 symptoms, most were female (78.1%), Latinx (87.5%, n = 28), and from mixed/mestizo racial background (59.4%, n = 19), while the average age was 48.6 years. Hence, indicating that long-term COVID-19 symptoms were prevalent among our community-based study population. Our results were consistent with previous studies that reported being 50+ years old and being from disadvantaged ethnic and socioeconomic groups as a risk factor, although comorbidities did not correlate with lack of COVID-19 symptom resolution, which could have been due to our small sample size of participants with lingering COVID-19 symptoms [28].

Finding high concordance (93.7%) between the POC results and the RBD/S (2 antigen excluding N) analysis for IgG outcomes in blood and a correlation between a positive POC test and increasing MFI multiplex measurements for IgG S and RBD (but not for N) in blood indicated that the RBD and S measurements (more so than N) were driving the overlapping results between the POC and multiplex outcome in our study population. Our and other studies have found N-based antibody levels to be lower and more variable (i.e., shorter half-life than S/RBD) [29, 30]. Hence, while we found that the POC test was an easy to use and reliable IgG vaccine-induced antibody measurement tool, the multiplex assay was more likely to pick up a positive sample and is more appropriate for serosurveys targeting and differentiating between infection- and vaccine-induced antibodies.

While the majority of SARS-CoV-2 serosurveys do not account for cross-reactivity between SARS-CoV-2 and hCoV measurements, we found that readout overlap for S between SARS-CoV-2 and OC43 could not be ruled out in our setting, indicating the need for further scrutiny in future serosurveys. This is particularly true because most individuals are thought to seroconvert for hCoVs during childhood [31–33] and variation in hCoV infection history has been proposed to induce protective immunity from COVID-19 [33–35].

In terms of methods, a current major challenge of serological analytics is (i) the application of validated across-plate normalization methods to pool outcomes from a large sample size,

and (ii) the determination of threshold values to reliably convert quantitative outcomes (MFIs) into qualitative results (positive/negative) [9]. We therefore validated a weighted-standard curve across-plate normalization method and two classification boundary methods for optimal qualitative serological assessments (one based on pre-defined positive and negative controls and one based on an alternate control group), across two isotypes (IgG and IgA) and two sample types (serum and saliva). As shown in the results, applying our methods resulted in the alignment of survey answers, POC results, and IgG-based serological outcomes with high classification accuracy, sensitivity and specificity in serum (**S2** and **S3 Tables** in S1 File) and saliva for almost all antigen combinations.

Specific to saliva-based serological analytics, (i) variation in salivary flow rate due to changes in circadian rhythm, stress, and sample collection method [36], and (ii) across sample variation in isotype specific-outcomes (IgA and IgG) due to inherent biological mechanisms (i.e., antibody source and half-life) are a major challenge [37–39]. Further, it is problematic to pool saliva-based serological outcomes across studies and identify appropriate controls since different saliva collection methods influence the composition and quality of the collected samples [40]. Here, we compared multiplex-based anti-SARS-CoV-2 IgG and IgA antibody measurements in matched serum and saliva samples. Our IgG-based serological outcomes in serum and saliva aligned, supporting the use of saliva as a less-invasive and accessible sample particularly among hesitant research participants. However, we found significant differences in antigen-specific IgA vs. IgG antibody levels, similarly to previous reports [7, 41]. This was likely because (i) the half-life of IgA is shorter compared to IgG, and (ii) mucosal and systemic IgA production are not synchronized [7, 37, 39, 41]. Further, the antigen-specific outcomes (MFI minus BSA for serum and transformed MFI for saliva) between serum and saliva did not correlate, even though the antigen-specific IgG seroprevalences aligned, underlining the importance of including appropriate controls and threshold calculation methods for final outcome comparisons.

The main limitation of our study was the restricted sample size. Hence, while our results generally align with previously published data, the statistical analyses and comparisons among infected individuals need to be confirmed among larger diverse populations. Further, most of our population carried anti-SAR-CoV-2 antibodies so subsequent statistical comparisons were restricted by the lack of negative outcomes.

In summary, this study successfully engaged marginalized MA communities and evaluated the impact of COVID-19 and vaccine uptake by implementing culturally sensitive recruitment methods and by giving appropriate study participant compensation in the form of immediate antibody results and adequate time and travel reimbursements.

We found a high vaccine uptake, and that about half of the participants were infected with SARS-CoV-2 within 2+ years of the beginning of the pandemic. We found that lingering COVID-19 symptoms were prevalent and impacted mostly middle-aged female Latinas, indicating continued need for public health attention despite high COVID-19 vaccine uptake. By comparison of matched blood and saliva samples, we found that saliva served as a reliable non-invasive alternative for IgG but not IgA antibody measurements, and we successfully adapted across plate normalization and classification boundary methods for optimal qualitative serological assessments. We also found that the bead-based multiplex assay had high overall sensitivity and specificity for blood samples and was more likely to pick up a positive sample than the POC. Overall, the bead-based multiplex assay was better suited for serosurveys targeting infection- and vaccine-induced antibodies compared to the less labor-intensive POC test and that hCoV cross-reactivity should be evaluated for reliable SARS-CoV-2 serosurvey results.

## Supporting information

**S1 Fig. Representative graphic of a three-dimensional classification boundary.** The graphic of a three-dimensional (3D) classification boundary graphic was based on training data covering anti-RBD, -S, and -N antibodies from confirmed positive and negative samples. S, Spike. RBD, Receptor Binding Protein. N, Nucleocapsid.
(TIFF)

**S2 Fig. Serum IgG outcome distribution in linear scale.** Linear scale dot plot with means and 95% confidence intervals (CI) of antigen-specific antibody measurements (MFI minus BSA and normalized across plates) for serum IgG SARS-CoV-2 and variants, along with human endemic coronaviruses OC43, HKU1, NL63, 229E. MFI, median fluorescence intensity. N, Nucleocapsid. S, Spike. RBD, Receptor Binding Protein. hCoV, human endemic coronaviruses. WT, wild-type (Wuhan).
(TIFF)

**S3 Fig. Analysis of samples that were non-concordant between the multiplex analysis and the point-of care test.** Comparison of samples that were positive for the multiplex assay and negative for the point-of care test (POC) test across all antigen combinations). (**A**) The average SARS-CoV-2 spike (S) median fluorescence intensity (MFI) of the non-concordant samples (n = 15) was lower compared to the concordant samples (n = 265, p = 0.0005). (**B**) The S-based OC43 measurements of the non-concordant samples were not significantly higher than the concordant samples (p = 0.12). Hence, while the multiplex assay exhibited high sensitivity and specificity for blood samples (**S2** and **S3 Tables** in S1 File) and was more likely to pick up a positive SARS-CoV-2 sample compared to the POC test, it was not more likely to pick up a positive OC43 sample among the non-concordant samples. The statistical outcomes may be affected by the unequal sample sizes (15 vs 265). MFI, median fluorescence intensity. S, Spike. Ns, non-significant (p>0.05). Non-conc., non-concordant samples (multiplex assay vs. POC test). Conc., concordant samples (sample that had the same qualitative outcome both with the multiplex assay and POC test). POC, point-of-care test. *** = p<0.001, Welch's t-test.
(TIFF)

**S4 Fig. Comparison of receptor binding protein-specific serum IgG and IgA outcomes.** Comparison of receptor binding protein (RBD)-specific serum IgG and IgA outcomes as median fluorescence intensity (MFI, mean and standard deviations) among the study participants (mean and standard deviations). **** = p<0.0001, Welch's t-test. MFI, median fluorescence intensity. RBD, Receptor Binding Protein.
(TIFF)

**S5 Fig. Comparison of spike-specific serum IgG and IgA outcomes.** Comparison of spike (S) protein-specific serum IgG and IgA outcomes (MFIs) among the study participants (mean and standard deviations). **** = p<0.0001, Welch's t-test. S, Spike Protein.
(TIFF)

**S6 Fig. Comparison of nucleocapsid-specific serum IgG and IgA outcomes.** Comparison of nucleocapsid (N) protein-specific serum IgG and IgA outcomes (MFIs) among the study participants (mean and standard deviations). **** = p<0.0001, Welch's t-test. N, Nucleocapsid Protein.
(TIFF)

**S7 Fig. Comparison of serum- and saliva-based serological outcomes.** Line up of saliva versus serum comparisons of antigen-specific outcomes (MFI minus BSA for serum and

transformed MFI for saliva [antigen- and isotype-specific MFI minus BSA, divided by total Ig, multiplied by 1000]), for anti-SARS-CoV-2 receptor binding domain (RBD; **A, B**), spike (S; **C, D**), and nucleocapsid (N; **E, F**) IgG (left column) and IgA (right column) antibody measurements. The outcomes between serum and saliva did not correlate for any antigen or isotype combination.
(TIF)

**S8 Fig. Comparison of serological outcomes from study versus control samples.** Comparison of study and control population by line up of saliva-based antigen-specific transformed MFI (antigen- and isotype-specific MFI minus BSA, divided by total Ig, and multiplied by 1000) for IgG (left column) and IgA (right column). For saliva IgG, the control sample population (sample collection method-matched samples from Kenya) always clusters in low MFI area and separate well from the study sample population (**A, C, E**), whereas for IgA the outcomes/MFIs from the control sample population overlap significantly with the study sample population for at least one antigen (**B, D, F**) and score higher maximum MFIs for RBD-specific outcomes (**B, D**). Hence, no saliva IgA percent seroprevalences could be calculated for the study samples based on these controls. S, Spike. RBD, Receptor Binding Protein. N, Nucleocapsid.
(TIF)

**S9 Fig. Comparison of receptor binding protein-specific saliva IgG and IgA serological outcomes.** Comparison of RBD-specific saliva IgG and IgA outcomes (transformed MFI = [raw MFI/total Ig]*1000) among the study participants (mean and standard deviations). **** = $p < 0.0001$, Welch's t-test. RBD, Receptor Binding Protein.
(TIFF)

**S10 Fig. Comparison of spike protein-specific saliva IgG and IgA serological outcomes.** Comparison of S-specific saliva IgG and IgA outcomes (transformed MFI = [raw MFI/total Ig]*1000) among the study participants (mean and standard deviations). **** = $p < 0.0001$, Welch's t-test. S, Spike.
(TIF)

**S11 Fig. Comparison of nucleocapsid protein-specific saliva IgG and IgA serological outcomes.** Comparison of nucleocapsid (N)-specific saliva IgG and IgA outcomes (transformed MFI = [raw MFI/total Ig]*1000) among the study participants (mean and standard deviations). **** = $p < 0.0001$, Welch's t-test. For N-specific outcomes in saliva, the IgA reads are higher compared to IgG. Whereas for RBD and S, the IgG reads in saliva are higher. Overall, the N-specific saliva IgG and IgA outcomes (transformed MFI) are lower than for RBD and S (i.e., lower overall MFI, see y-axis comparison between S8, S9, and S10 Figs). N, Nucleocapsid.
(TIFF)

**S1 File. This is a template of the data collection survey utilized for this study.**
(PDF)

**S2 File. This document contains supplemental information on the study and associated methods as referenced throughout the manuscript.**
(PDF)

**S1 Dataset. This document contains the raw data utilized for the study analysis, covering both the survey results and serological outcomes.**
(XLSX)

## Acknowledgments

We thank the study participants and their families for giving us their valuable time and attention, they are the core of this study. We thank Centro Inc. (Centro las Americas) in Worcester, MA, Dr. Juan A. Gomez, and the amazing team on the ground for partnering with us on this study. We thank Pamela Suprenant from the Central MA YMCA, Pastors Richard (Richie) and Elizabeth Gonzalez from Net of Compassion, the Parish of Saint John the Guardian of Our Lady in Clinton, MA and their welcoming congregation, and the UMass Chan iCELS team for inviting us to recruit and engage their communities. This study would not have been possible without the essential support of these individuals and their institutions. Use of data in this manuscript was approved by the NIST Research Protections Office under study number ITL-2020-0257. Certain commercial equipment, instruments, software, or materials are identified herein to specify the experimental procedure. Such identification does not imply recommendation or endorsement by NIST nor does it imply that the materials or equipment identified are necessarily the best available for the purpose.

## Author Contributions

**Conceptualization:** Raquel A. Binder, Ann M. Moormann.

**Formal analysis:** Raquel A. Binder, Prajakta Bedekar, Paul N. Patrone.

**Funding acquisition:** Ann M. Moormann.

**Investigation:** Raquel A. Binder, Angela M. Matta, Prajakta Bedekar, Paul N. Patrone, Boaz Odwar, Jennifer Batista, Sarah N. Forrester, Heidi K. Leftwich, Lisa A. Cavacini, Ann M. Moormann.

**Methodology:** Angela M. Matta, Catherine S. Forconi, Cliff I. Oduor, Prajakta Bedekar, Paul N. Patrone, Anthony J. Kearsley, Lisa A. Cavacini, Ann M. Moormann.

**Project administration:** Catherine S. Forconi, Anthony J. Kearsley, Boaz Odwar, Jennifer Batista.

**Resources:** Cliff I. Oduor, Lisa A. Cavacini, Ann M. Moormann.

**Supervision:** Raquel A. Binder, Ann M. Moormann.

**Writing – original draft:** Raquel A. Binder.

**Writing – review & editing:** Raquel A. Binder, Angela M. Matta, Catherine S. Forconi, Cliff I. Oduor, Prajakta Bedekar, Paul N. Patrone, Anthony J. Kearsley, Boaz Odwar, Jennifer Batista, Sarah N. Forrester, Heidi K. Leftwich, Lisa A. Cavacini, Ann M. Moormann.

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
