## [Decision Letter · Decision Letter 0]

9 May 2024

PONE-D-24-10270Minding the margins: Evaluating the impact of COVID-19 among Latinx and Black communities with optimal qualitative serological assessment tools.PLOS ONE

Dear Dr. Binder,

Thank you for submitting your manuscript to PLOS ONE. After careful consideration, we feel that it has merit but does not fully meet PLOS ONE’s publication criteria as it currently stands. Therefore, we invite you to submit a revised version of the manuscript that addresses the points raised during the review process.

We look forward to receiving your revised manuscript.

Kind regards,

Andrey I Egorov

Academic Editor

PLOS ONE

Journal Requirements:

"This work was supported through the National Institutes of Health, NCI Serological Sciences Network (U01 CA261276), UMass Chan COVID-19 pandemic research fund, MassCPR Evergrande Award, and the National Center for Advancing Translational Sciences, National Institutes of Health, through Grant KL2-TR001455. The content is solely the responsibility of the authors and does not necessarily represent the official views of the NIH."

"None of the authors declare a conflict of interest. "

Reviewers' comments:

Reviewer's Responses to Questions

**Comments to the Author**

1. Is the manuscript technically sound, and do the data support the conclusions?

Reviewer #1: Yes

Reviewer #2: Yes

2. Has the statistical analysis been performed appropriately and rigorously? 

Reviewer #1: Yes

Reviewer #2: Yes

3. Have the authors made all data underlying the findings in their manuscript fully available?

Reviewer #1: Yes

Reviewer #2: Yes

4. Is the manuscript presented in an intelligible fashion and written in standard English?

Reviewer #1: Yes

Reviewer #2: Yes

5. Review Comments to the Author

Reviewer #1: Reviewer comments:

The authors have made a detailed qualitative serological assessment to investigate the impact of COVID-19 among the Latinx and Black communities. After going through the manuscript, I have found that the parameters for participant recruitment, sample collection, and sample processing had been performed meticulously (as the study involved culturally sensitive-marginalized communities) and were lucidly justified in the manuscript. The following of concerned protocols regarding the handling of human samples and due approval of the concerned authority/ethical committee is appreciated. The survey was performed to answer mainly three important investigations, i.e. (i) the impact of COVID-19 and vaccine uptake among marginalized communities, (ii) the utility of using saliva for serosurveys, (iii) a comparison of the utility of a bead-based multiplex assay vs. a point-of-care (POC) test for SARS-CoV-2 antibody measurements, and (iv) demonstration of the benefit of developing and using classification boundary methods for optimal interpretation of serological assays. The main research techniques used in the study were: Multiplex Luminex assay, POC test, across-plate normalization, and Qualitative assessment using statistical analysis.

The following few points need to be highlighted by the authors to aid the readers in perceiving the statements made thereof.

Methods (Qualitative serological assessment)

The authors have reflected on the use of alternative control samples from a Kenyan study for qualitative serological assessment which is elaborated in the supplementary information. However, I feel the need to cite the reference of that particular Kenyan study in the methods section itself for the ease of the readers.

Results

Demographics and Vaccine/Infection History: The authors have elaboratively discussed the population demographics and the history of COVID-19 vaccinations in the Latinx and Black communities. As the study is highly based on the self-declaration of health status by individuals of these communities with a very limited sample size, I feel the need to mention/compare the demographics with previous studies of COVID-19 on the Latinx and Black communities (as some more references are available).

Blood-based SARS-CoV-2 Antibodies test:

The POC test covered both SARS-CoV-2 N and S antigens, while the multiplex assay allowed measuring the presence of antibodies based on individual antigens and therefore distinguishing between vaccine and infection-induced antibodies. The POC test was found reliable for detecting the RBD/S antibodies measured by the multiplex assay. This section needs no further changes.

Discussion:

As mentioned by the authors, the antigen-specific outcomes between serum and saliva did not correlate, even though the antigen-specific IgG seroprevalences aligned. However, the authors have been able to ascertain the use of saliva as a less-invasive and accessible sample. Furthermore, this research highlights the need of hCoV cross-reactivity to be evaluated for reliable SARS-CoV-2 serosurvey results.

Final comment: This research investigation is of potential importance as it makes important remarks on the impact of the COVID-19 disease among marginalized ethnic races using serological assessment tools and also devices some qualitative assessment parameters that will serve as effective investigation methods for further large-scale population-based surveys. The authors were able to answer the questions addressed initially in the investigation. No significant grammatical/English corrections were noticed in the manuscript. The manuscript is suitable for publication with very minor changes as mentioned above.

Reviewer #2: The authors show the feasibility of using saliva and matching blood samples to determine SARS-CoV-2 antibody seroprevalence among a predominantly female and Hispanic population. They found good correlation between self-reported vaccination status (most were vaccinated against COVID) and a POC SARS-CoV-2 antibody device they employed on-site during the study visit and a multiplex assay that was used to test both blood and saliva for SARS-CoV-2-specific and endemic coronavirus IgG and IgA with the exception for salivary IgA.

My main comment is that the authors should address the discrepancy they see between prior self-reported COVID-19 infection and discrepant classification by blood and maybe also saliva test. ~40% of participants reported having had COVID-19 (test confirmed, n = 121 participants) but about 50 of them did not test positive for anti-N IgG. Similarly, ~60 out of ~160 participants who reported not having had COVID-19 do test positive for anti-N IgG. It is a bit odd to stratify Table 1 by this outcome and then not address it.

Another comment is that while normalization of saliva antibody signals with total Ig CONCENTRATION is often used, normalization with MFI values only works if both, the pathogen-specific signal and the total Ig MFI signal are within the linear range of the assay. The non-transformed (raw) data is not provided.

Some minor comments:

Line 235 “[…] indicating past infection rather than vaccination and mirrored self- reported exposures.”

The percentage of self-reported infection may be similar to the percentage of anti-N positive blood samples but Table 1 appears to suggest that only ~50% of participants who reported a prior SARS-CoV-2 positive test also tested anti-N positive. Conversely, ~ one third of those who did not report a prior SARS-CoV-2 positive test had a anti-N positive result. The authors may want to acknowledge this here or omit the “mirroring” statement if addressed in the discussion.

Line 242 As for SARS-CoV-2 variants, the delta variant had the most abundant antibodies among our study population, see Fig 1.

Higher MFI does not necessarily mean most abundant antibody. MFI signals are influenced by a lot of factors including quality of the antigen, orientation of immunogenic (antibody-binding sites) regions of the antigen on the bead, antigen density on the bead, orientation of the bound antibody on the bead, etc. The study took place during the Omicron wave and, as the authors mentioned, most participants were vaccinated (i.e., “primed” with the Wuhan strain). Consider acknowledging that MFI signal strength alone does not mean that most participants were infected with Delta or revise sentence accordingly.

Line 263 “[…] indicating that the POC test was reliably detecting the RBD/S antibodies measured by the multiplex assay.”

Are the authors trying to say that the POC assay does only classify blood samples with high/higher anti-S MFI as positive? Please clarify. Also, a brief description about what is meant with 2/3 antigen positivity (S/N; RBD/N, etc.) would be helpful. I.e., positive for IgG to all antigens or at least one of them, etc.? It might also be helpful to indicate the multiplex assay outcome (for one of the algorithms?) in Figure 2, e.g., coloring the dots according to the multiplex assay result or by using different (larger) symbols according to result and/or adding a threshold for S MFI / RBD MFI / N MFI.

Line 266 “negative for the POC test across all antigen combinations”

Does the POC test detect antibodies against multiple antigens?

Line 300 While anti-SARS-CoV-2 IgA has been shown to resolve faster than IgG

Maybe “wane” or “decline” would be the better word?

Line 350 “the first COVID-19 case in MA was confirmed on Feb 1st 2022”

Should this be 2020?

Line 363 groups as a risk factor, although and comorbidities did not correlate with lack of COVID-9 symptom resolution,

Grammar? Delete “and” I assume.

6. PLOS authors have the option to publish the peer review history of their article (what does this mean?). If published, this will include your full peer review and any attached files.

Reviewer #1: No

Reviewer #2: No

---

## [Author Response · Author response to Decision Letter 0]

23 Jun 2024

Thank you very much for the thoughtful review of the manuscript titled “Minding the margins: Evaluating the impact of COVID-19 among Latinx and Black communities with optimal qualitative serological assessment tools”. We greatly appreciate the reviewers’ comments which greatly improved the manuscript. Please see below our point-by-point responses in blue and italics. 

Reviewer #1: 

Summary:

---

## [Editor Report · Decision Letter 1]

4 Jul 2024

PONE-D-24-10270R1Minding the margins: Evaluating the impact of COVID-19 among Latinx and Black communities with optimal qualitative serological assessment tools.PLOS ONE

Dear Dr. Binder,

Thank you for submitting your manuscript to PLOS ONE. After careful consideration, we feel that it has merit but does not fully meet PLOS ONE’s publication criteria as it currently stands. Therefore, we invite you to submit a revised version of the manuscript that addresses the points raised during the review process.

Please address the following minor issues: Re-number supplemental tables and figures. In the current version, Table S5 is the first supplemental table referenced in the Supplemental methods while Figure S11 is the first supplemental figure referenced in the text on page 12.

Please check the names of companies and products. It should be Sino Biological rather than Sino Biology in Table S5.  

Please make sure to use consistent and logical terminology and definitions throughout the text. For example, please correct statements in Lines 98-101 and in 2^nd^ paragraph in the Supplemental Methods. The POC test detects IgG and IgM antibodies to S and N antigens (anti-SARS-CoV-2 antibodies), rather than anti-immunoglobulin antibodies in blood samples. Also, this test does not detect S and N antigens in blood.

Lines 358 – 361. It appears that previously infected (“exposed”) individuals include those who tested seropositive to the N antigen as well as those seronegative individuals who had been diagnosed with SARS-COV-2 infection (Table 1).  

Lines 395 and 450. No data to compare sensitivity values of Luminex and POC tests is provided in the paper. Please provide such data or remove statements about Luminex assay being more sensitive.

Supplemental Information, Multiplex Assay – Saliva: Please confirm that undiluted saliva samples were tested for total IgG and IgA. The concentration of total Ig in saliva is so high that it would be reasonable to use highly diluted saliva in this analysis.

We look forward to receiving your revised manuscript.

Kind regards,

Andrey I Egorov

Academic Editor

PLOS ONE
---

## [Author Response · Author response to Decision Letter 1]

5 Jul 2024

Thank you for the detailed review and additional feedback for the manuscript titled “Minding the margins: Evaluating the impact of COVID-19 among Latinx and Black communities with optimal qualitative serological assessment tools”. Please see below our point-by-point responses in blue. 

1. Re-number supplemental tables and figures. In the current version, Table S5 is the first supplemental table referenced in the Supplemental methods while Figure S11 is the first supplemental figure referenced in the text on page 12. We streamlined the numbering for the supplemental figures. S11 was the second supplemental figure to be listed and it is now labeled S2, while all the other supplemental figure numberings have been adjusted. The reason that Table S5 is the first supplemental table referenced in the supplemental methods is because Table S1-S4 are listed first in the main manuscript. 

2. Please check the names of companies and products. It should be Sino Biological rather than Sino Biology in Table S5. Thank you for catching that. The company name has been corrected and the other ones verified. 

3. Please make sure to use consistent and logical terminology and definitions throughout the text. For example, please correct statements in Lines 98-101 and in 2nd paragraph in the Supplemental Methods. The POC test detects IgG and IgM antibodies to S and N antigens (anti-SARS-CoV-2 antibodies), rather than anti-immunoglobulin antibodies in blood samples. Also, this test does not detect S and N antigens in blood. The POC test description in the main text and supplemental method has been corrected and adapted. 

4. Lines 358 – 361. It appears that previously infected (“exposed”) individuals include those who tested seropositive to the N antigen as well as those seronegative individuals who had been diagnosed with SARS-COV-2 infection (Table 1). Yes, not all those who self-reported a positive test were also positive for N antibodies. See lines 372-88. “The percent of self-reported infections aligned with the N-based IgG seroprevalence outcomes in serum (N: 49.9 %) and saliva (N: 48.0 %). Hence, within two years and four months (the first COVID-19 case in MA was confirmed on Feb 1st, 2020 (24) and the study recruitment ended in July of 2022) about half our diverse study population had been exposed to SARS-CoV-2. While the percent of self-reported infections aligned with the N-based IgG serum and saliva seroprevalence results, we found that reporting a positive SARS-CoV-2 test was not necessarily linked to the presence of anti-N IgG antibodies (Table 1). This is likely due to the relatively short half-life of anti-N IgG antibodies. Others have found that anti-N antibodies start declining within one month post-positive PCR test with over half the study population testing seronegative within 6-7 months.(25, 26) Since our study was implemented 2+ years after the first local COVID-19 case and since antibody levels may range across individuals, it is likely that the antibody levels had dropped below the detection limits by the time we collected and screened the blood samples of these individuals. Similarly, we found that participants who did not report a confirmatory test were positive for anti-N IgG antibodies. Given that 4% to 41% of SARS-CoV-2 infections may be asymptomatic (27), it is likely that these individuals may have been infected but were not aware or did not seek testing.” 

5. Lines 395 and 450. No data to compare sensitivity values of Luminex and POC tests is provided in the paper. Please provide such data or remove statements about Luminex assay being more sensitive. The sensitivity and specificity for the blood-based Luminex analyses was computed and listed in S2 Table and S3 Table in supplemental material but not for the POC. The sentences have been amended accordingly. See lines 303-306, 529-532 and 473/4. 

6. Supplemental Information, Multiplex Assay – Saliva: Please confirm that undiluted saliva samples were tested for total IgG and IgA. The concentration of total Ig in saliva is so high that it would be reasonable to use highly diluted saliva in this analysis. Yes, as indicated in the first paragraph of the “Multiplex Luminex Assay – Saliva” section in the supplemental material, we used undiluted saliva to screen for anti-SARS-CoV-2 antibodies. While the total Ig in saliva was high (as expected/previously described), the MFI for the CoV-specific antibodies had a wide range (some low depending on the antigen measured), which led us to use undiluted saliva for the screening.

---

## [Editor Report · Decision Letter 2]

9 Jul 2024

Minding the margins: Evaluating the impact of COVID-19 among Latinx and Black communities with optimal qualitative serological assessment tools.

PONE-D-24-10270R2

Dear Dr. Binder,

We’re pleased to inform you that your manuscript has been judged scientifically suitable for publication and will be formally accepted for publication once it meets all outstanding technical requirements.

Kind regards,

Andrey I Egorov

Academic Editor

PLOS ONE
---

## [Editor Report · Acceptance letter]

15 Jul 2024

PONE-D-24-10270R2 

PLOS ONE

Dear Dr. Binder, 

I'm pleased to inform you that your manuscript has been deemed suitable for publication in PLOS ONE. Congratulations! Your manuscript is now being handed over to our production team.

Kind regards, 

on behalf of

Dr. Andrey I Egorov 

Academic Editor

PLOS ONE